# Trophic Position Stability of Benthic Organisms in a Changing Food Web of an Arctic Fjord Under the Pressure of an Invasive Predatory Snow Crab, *Chionoecetes opilio*

**DOI:** 10.3390/biology13110874

**Published:** 2024-10-28

**Authors:** Anna K. Zalota, Polina Yu. Dgebuadze, Alexander D. Kiselev, Margarita V. Chikina, Alexey A. Udalov, Daria V. Kondar, Alexey V. Mishin, Sergey M. Tsurikov

**Affiliations:** 1Shirshov Institute of Oceanology, Russian Academy of Sciences, 117997 Moscow, Russia; ad-kiselev@mail.ru (A.D.K.); chikina@ocean.ru (M.V.C.); aludal@mail.ru (A.A.U.); kondaria@gmail.com (D.V.K.); mishin-aleksej@mail.ru (A.V.M.); 2A.N. Severtsov Institute of Ecology and Evolution, Russian Academy of Sciences, 119071 Moscow, Russia; p.dgebuadze@gmail.com (P.Y.D.); smtsurikov@rambler.ru (S.M.T.)

**Keywords:** food web structure, invasive species, food source, stable isotope analysis, Arctic marine ecosystem, benthos

## Abstract

Impacts caused by alien species are often obscured due to other biotic and abiotic stressors impacting ecosystems worldwide. The introduction of a new predator into a previously undisturbed ecosystem is a rare opportunity to observe such impacts in its pure form. The invasion of the snow crab, *Chionoecetes opilio*, in Blagopoluchiya Bay of the Kara Sea is a rare natural experiment. The crab settled in the bay in 2014 in high numbers, and up to the present day, this one cohort lives and grows in the bay, causing changes in the taxonomic structure, abundance, and biomass of local benthic communities. The results of stable isotope analysis revealed the stability of the trophic positions and sources of organic material for the most common benthic species as the invasion progressed. As the macrobenthic community of the bay changed, any changes in the prey items of the consumers were within the same trophic level. The food web structure changed, where secondary consumers became more numerous, and the proportion of deposit and suspension feeders decreased. The remaining benthic organisms are less available as food items for the snow crab, which may result in starvation and further decline of the invasive crab population.

## 1. Introduction

It is now hard to find a place on Earth that is not invaded by non-indigenous species [1,2,3,4]. Moreover, both land and water environments are affected by invasive species that have negative impacts on local ecosystems [5]. The spectrum of negative impacts of invasions on ecosystems is very diverse. Invasions can lead to the displacement of native species due to changes in the structure and functioning of native ecosystems; they can be environmental engineers, alter the physical properties of the habitat, and bring along parasites and pathogens [2,6]. Marine environments are often thought to be more resilient; however, this resilience has limits [7,8]. Marine invasive species can positively impact ecosystem services by establishing new commercial fishing, recreation, and tourism services, such as the king crab in the Barents Sea, which has created a large commercial fishing and tourism industry in both Russia and Norway [5,9,10,11]. The negative impacts are often hard to distinguish from other stressors affecting an ecosystem (such as climate change, habitat degradation, etc.). It is generally assumed that the introduction of new species is harmful, and further spread should be controlled [12].

Historically, the Arctic was considered a low-risk region for biological invasions due to harsh environmental conditions and limited human activity [13]. However, climatic changes affecting ice cover dynamics and increasing human presence in the region have resulted in the introduction of various invasive species [14,15]. Until recently, the Arctic Kara Sea was one such place with low human impact and no non-indigenous species in its benthic ecosystem [16]. In fact, the distribution and species composition of the Kara Sea benthic communities observed in the 1920s and 1930s remained notably stable until the 2010s [17,18,19,20,21]. Recently, the Kara Sea has experienced the general Arctic trend of a slowdown in sea ice formation in autumn and its earlier destruction in spring–early summer [22]. This has facilitated the introduction of an invasive crab by providing enough time and suitable conditions for crab larvae, which inflow from the Barents Sea and recently hatch in the Kara Sea, to develop and settle [23].

The snow crab, *Chionoecetes opilio* was first found in the Barents Sea in 1996 [24]. To date, there is no direct evidence of its negative impact on the Barents Sea benthic communities, although such a large predator is expected to have adverse effects, especially with further growth and expansion of its population [25,26]. After rapid population growth and spread, it entered the Kara Sea, where it was found in 2012 [27]. By 2014, juvenile snow crabs had spread throughout the western part of the sea and in the bays of the eastern coast of the Novaya Zemlya Archipelago [28]. It has now entered the eastern part and established a reproducing population in the western part of the Kara Sea [23,29]. Since the invasion of this crab, sudden and dramatic changes in the benthic fauna of the southwestern part of the Kara Sea have been observed [16,30,31]. The most pronounced changes occurred in the benthic communities of Blagopoluchiya Bay, the northern fjord of the Novaya Zemlya Archipelago [16].

Snow crabs settled in Blagopoluchiya Bay in 2014 [16]. In 2016, a large number of small (carapace width CW ~15 mm) juvenile crabs were caught in trawl samples, and this single cohort grew throughout the years without any new settlements [16,28,32]. By 2018, crabs grew (CW ~30 mm), and their abundance was more than 20 crabs per 10 m^2^. In 2020, their number crashed (fewer than ten crabs per 10 m^2^) and further decreased in 2022 [16].

Blagopoluchiya Bay’s benthic communities were studied before and during the invasion [16,33]. Changes in benthic communities occurred at various levels and were first observed after 2018 when the crab reached a large abundance and size that could feed on larger prey, such as mollusks and brittle stars. While changes in macrobenthos were relatively weak, changes in megabenthos were dramatic. By 2022, not a single specimen of previously dominant ophiuroids was found in the bay [16]. No consistent trends in abiotic parameters, such as ice melting, turbidity, temperature, and nutrients, were observed during the observation period in the northwestern part of the sea along the coast of the Novaya Zemlya Archipelago and Blagopoluchiya Bay [16]. It has been shown that the changes in benthic communities of the bay cannot be attributed to environmental factors; they are the result of the establishment and development of the snow crab population [16].

Blagopoluchiya Bay is a long and narrow (12 by 5 km) Arctic fjord with a deep Inner Basin (180–200 m deep) and a Sill (40 m deep) separating it from the outer part of the fjord. Climate conditions in Blagopoluchiya Bay are severe, and the bay is covered with ice most of the year (from the end of October to June). Freshwater input comes from two rivers, bringing high volumes of terrestrial mineral material with them [33,34]. Fjord ecosystems may be highly sensitive to various stressors due to the lower functional redundancy of macrofauna compared to the open sea [35], which makes Blagopoluchiya Bay a unique model fjord to study the impact of an invasive predator on an otherwise pristine ecosystem.

The impact of invasive species on native communities is a highly discussed issue in invasion ecology [2]. Biological invasions also provide a unique opportunity to study fundamental processes in populations, communities, ecosystems, and the evolution of many animal groups [6]. Trophic interactions are one of the main types of relationships between animals in an ecosystem, and the study of the structure and functioning of food webs allows us to assess the stability of an ecosystem [1,36]. Changes in food webs can ultimately affect all species inhabiting an ecosystem [37]. The impact level of an invader may strongly depend on the position and niche it occupies [38,39]. Negative impacts from predation and competition for food are often the most severe [37]. In the literature, the snow crab is described as a predator or omnivorous species, but with the predominance of animal food in its diet [40,41,42,43]. Before the snow crab invasion, there were no large predators in the benthic communities of the Kara Sea. There are no potential predators for the snow crab except for the crab itself.

The theory of trophic dynamics allows us to model the energy pathways in an ecosystem in the form of discrete trophic levels (TLs) [44]. Trophic position (TP), unlike TL, is measured on a continuous scale, which accounts for the omnivory of species, which is especially true in marine ecosystems [45]. Individual species TP and overall community trophic structure provide powerful means to assess the resilience and sensitivity of food webs under stress [46]. Stable isotope analysis is a widely used tool to assess the trophic position and the source of organic material in marine ecosystems [47,48,49]. The carbon isotope ratio changes little across trophic levels and is used to determine the source of organic material (source of primary production) [50,51]. The nitrogen isotope ratio is typically enriched by 3–4‰ between the prey and the consumer, which allows us to infer their trophic position [52,53]. Unlike time- and resource-consuming classical studies of organisms’ feeding habits (such as stomach and excrement content analyses that mostly show ingestion data), stable isotope analysis allows us to obtain general information about species’ feeding habits in terms of their trophic position. A comparison of the results from stable isotope analysis with available literature data can shed light on the food sources of many understudied species.

Benthic invertebrates are often assigned to four feeding modes (or feeding types), mostly based on the morphology of their mouthparts and their positioning on the seafloor [54]. Suspension feeders capture food particles from the water column [55,56]. Deposit feeders are subdivided into surface and subsurface deposit feeders, which acquire food from deposited material on the surface or deeper sediment horizons [57]. Predators/scavengers actively hunt their prey or scavenge on dead organisms [56,58]. In most cases, stable isotope analysis is not sensitive enough to differentiate between predators that feed on live or dead animals. Furthermore, bulk and even compound-specific stable isotope analyses do not show significantly different results between surface and subsurface deposit feeders [48,59,60]. In all feeding mode groups, species exhibit high variability in their preferences of primary production sources and their trophic levels [60,61]. This variability can be explained by the high diversity of genera, sizes, selectivity, and mobility of species belonging to these broad feeding modes [62]. More detailed analyses of traits and diet (such as fatty acid and compound-specific analyses) could help explain such variations. However, for most benthic invertebrate species, there is little information about their diet and plasticity under different conditions. Stable isotope analysis of organisms in a marine food web with different feeding modes could be the first step toward such an understanding and reveal their general response to the changes in food availability.

As the invasion of snow crabs in Blagopoluchiya Bay has resulted in changes in species composition and abundance of benthic organisms [16], and this direct influence is not obscured by anthropogenic interference and climate change, this bay is a useful model system to study different aspects of negative impacts by an invasive species. In particular, this study looks at the trophic structure of the community and individual species’ feeding preferences in terms of their trophic position. For each studied species, we use literature data on its feeding habits in other habitats and compare it with its trophic position in Blagopoluchiya, which we obtain using stable isotope analysis. We are interested in seeing if species alter their feeding habits (feeding on a different trophic level or from a different source of primary production).

## 2. Materials and Methods

### 2.1. Sample Collection

Samples for stable isotope analysis were collected in September–October 2018, 2020, and 2022 during R/V cruises of the Shirshov Institute of Oceanology within the framework of a long-term study, “Ecosystems of the Siberian Arctic Seas”. Samples were collected at two monitoring stations representing the two most common habitats of the bay, and these stations have been sampled several times throughout the years. The Inner Basin station is located in the deepest inner part of the bay at 150–170 m, with high levels of inorganic matter sedimentation; the shallower Sill station, located near the exit of the bay, is at a depth of 63–72 m (Figure 1) and has lower levels of sedimentation, being more connected to the open Kara Sea. The two stations have different benthic communities, with a greater proportion of suspension feeders at the Sill station and a larger proportion of deposit feeders at the Inner Basin station [16].

In all years, benthic mega- and macrofauna samples were collected using a Van Veen grab (0.1 m^2^ sampling area) and a Sigsbee trawl with a frame size of 150 cm × 35 cm (with an inner mesh of 0.5 cm). In addition, in 2022, we collected sediment up to a depth of ¼ of a box corer (0.25 m^2^ sample area). Van Veen grab and box corer samples were washed through a 0.5 mm mesh sieve, while trawl samples were washed through a set of 5 mm and 1 mm mesh sieves.

The top 1–2 cm of the sediments were scraped from the surface of the box corer to collect meiobenthic fauna in 2022. Meiobenthic organisms were extracted by sequentially sieving sediment through 250, 125, 63, and 40 μm mesh sieves using filtered seawater (GF/F filter 0.7 μm). These subsamples were then centrifuged at 1000 rpm for 30 min and washed in distilled water to separate living organisms from the deposits. At least 200 specimens of live nematodes and 50–70 specimens of harpacticoid copepods were selected under a stereomicroscope to obtain a minimum of 50 μg dry-weight samples for analysis.

In 2020 and 2022, zooplankton samples were collected using a Bongo plankton net (opening diameter of 60 cm and mesh size of 500 μm) dragged from the bottom to the surface.

The material was identified to an accessible taxonomic level, depending on the specialists present during the sampling year, and the most abundant and representative species were selected for analysis. Muscles were collected from animals (crabs, shrimps, and fish) when possible. The foot was selected from large mollusks, which were washed with seawater and cleaned with a scalpel. Ambulacral feet were obtained from the sea stars. Large polychaetes were cleaned of external structures and digestive organs (squeezed out). Small organisms were cleaned of calcified body parts when possible or sampled whole. We aimed to collect at least five replicates for each species; however, often, there was not enough material, and fewer samples were obtained.

Samples of macrophyte algae were collected in 2020 and 2022 from trawl samples and the washout piles on the eastern shore opposite the Inner Basin station. All collected samples were detached and in various stages of decay. The most wholesome parts of the plants were selected, washed in seawater, and scraped with a scalpel to remove epifauna. Since there are no known species-specific primary consumers in the Blagopoluchiya Bay benthic fauna, the species composition of macrophytes was not identified. In addition, most of the sampled material consisted of fragments of partially decomposed algae, which would be difficult to identify.

The samples were dried at 70 °C for five days. Specimens whose calcified body parts could not be removed (ophiuroids, soft corals *Gersemia* sp., small bivalves *Ennucula tenuis*, *Portlandia arctica*, and *Thyasira* sp.) were divided into two subsamples. One subsample was acidified with 1N HCL to remove inorganic carbon and then re-dried. Both acidified and non-acidified subsamples were analyzed (see Appendix A). Carbon values were obtained from acidified samples, while nitrogen values were obtained from non-acidified samples to minimize the error associated with acidification [63]. The dried material was then crushed using a mortar and pestle and wrapped in tin foil containing 200–500 μg of animal material or approximately 1500 μg of plant material. Isotope analysis was performed with a Thermo Delta V Plus isotopic mass spectrometer and Thermo Flash 1112 Elemental Analyzer at the Centre for Collective Use at the Severtsov Institute of Ecology and Evolution, RAS. The ratio of stable isotopes (^13^C/^12^C and ^15^N/^14^N) is presented in per mil units (δ, ‰) of deviation from international standards (Vienna PeeDee Belemnite (VPDB) for δ^13^C and atmospheric N_2_ for δ^15^N). The analytical error in determining the isotope composition (SD in the laboratory standard analysis of protein B2155, *n* = 6–8) did not exceed 0.2‰. The δ^13^C values of animals with a mass C/N ratio higher than four were corrected for lipid concentration using the mathematical equation proposed by Post et al. [64] (see Appendix A). Mean δ^13^C and δ^15^N values ± standard deviation (SD) are reported in Table 1, Table 2 and Table 3.

All organisms were assigned to three broad feeding modes based on available literature (Table 4). These feeding modes reflect where the food is obtained, not the preference for food items. Deposit feeders are organisms that collect food from the surface or subsurface of sediments, mainly by ingesting sediments; suspension feeders collect food from the water column; and active predators are mobile species that actively scavenge on dead or prey on live animals [65]. Isotopic data were used to calculate the trophic positions of these species, which can exhibit varying levels of omnivory and carnivory within the same feeding mode guild.

### 2.2. Data Analysis

The isotopic trophic position of consumers was calculated using the full (two-baseline) parametric model for multiple species in the tRophicPosition package 0.7.7 [66], using RStudio 2024.04.01 [67,68]. Default settings of discrimination factors were used, where ΔN is set to 3.4 ± 0.98 and ΔC is set to 0.39 ± 1.3, based on Post [53]. The two-baseline approach includes calculating an alpha (α), representing the proportion of N derived from baseline 1 [53,69]. The trophic position of a consumer (TPc) is calculated using the following formula:TPc = λ + (δ^15^N_c_ − (δ^15^N_b1_ × α + δ^15^N_b2_ × (1 − α)))/ΔN
and,
α = ((δ^13^C_b2_ − (δ^13^C_c_ + ΔC))/(TPc − λ))/(δ^13^C_b2_ + δ^13^C_b1_)
where δ^13^C_c_ and δ^15^N_c_ are isotopic values of the consumer, δ^13^C_b1_ and δ^15^N_b1_ of benthic macrophytes, and δ^13^C_b2_ and δ^15^N_b2_ of phytoplankton algae, which were selected as the baseline, and their trophic level set to one (λ = 1). Feeding on reprocessed organic matter and macrophytes can raise carbon isotopic ratios of consumers while feeding on fresher pelagic primary production leads to lower δ^13^C values. This is reflected in the alphas calculated using the full TP model [49,70,71]. Therefore, an alpha of 1 suggests that all of the organic carbon of the consumer is derived from benthic macroalgae sources, while an alpha of 0 indicates that phytoplankton is the primary source.

To describe the baseline (trophic level 1, primary producers (PP)) of the trophic web, we used the most abundant and large macrophyte species, kelp and Fucus, as benthic primary producers. In both 2020 and 2022, these species had similar δ^13^C and δ^15^N value ranges (Table 2). These are long-living species; therefore, we used values from both years and applied them to all data. For pelagic primary producers, we calculated phytoplankton isotopic values from *Calanus* spp. (which feed predominantly on phytoplankton [72])a by subtracting a trophic step discrimination factor of 1‰ from δ^13^C values and 3.4‰ from δ^15^N values (Table 2 and Table 3) [51,73]. This manuscript’s Results and Discussion sections further discuss these baselines’ choices. To assess the errors associated with different baseline combinations, this model was run multiple times using baseline species collected only in 2020 or 2022 for these respective years (baseline species were not sampled in 2018) and for the bulk baseline from both years. A non-parametric Kruskal–Wallis rank sum test was used to check the differences between TP and alpha values obtained using different baselines, followed by Dunn’s multiple comparison test with *p*-values adjusted using the Holm method.

The discrimination factors can vary between species, different tissues sampled, and at different trophic levels [47,74,75]. It is difficult to account for the diet and tissue variability since experimental data is scarce. The rescaled TPs were calculated using a method proposed by [47] to account for the variability of discrimination factors at different trophic levels. The bulk average δ^15^N values of macrophyte and phytoplankton algae were used as the baseline for this model.

## 3. Results

Overall, 290 samples from 47 animal taxa (Table 1) were analyzed using stable isotope analysis. The δ^13^C values of animal samples ranged across 10‰, from −26.0 to −16.7‰, and δ^15^N values across 14‰, from 6.9 to 20.2‰.

In addition, 23 samples of macrophyte algae were analyzed from 2020 and 2022 (Table 2). The δ^13^C values of macrophyte algae ranged across more than 20‰, from −37 to −15.5‰, and δ^15^N values across 4‰, from 2.3 to 6.7‰. The macrophyte algae were generally not identified at the species level but were differentiated based on a broader taxon. Red algae were present in small abundance and, although not identified, were differentiated into five different taxonomic groups. Their δ^13^C values ranged from −37 to −19.8‰ and from 2.3 to 6.7‰ in δ^15^N, with three samples concentrating around −36‰ and the rest around −22‰ in δ^13^C values. Brown algae, kelp, and Fucus had similar narrow δ^15^N value ranges in both 2020 and 2022 (from 3.5 to 5.9‰) but a broad δ^13^C value range (from −23.2 to −15.5‰). The obtained phytoplankton values slightly differed between the years (by approximately 1.5‰ in δ^13^C values and 2‰ in δ^15^N values). However, their δ^13^C values differed by more than 2‰, and the δ^15^N values were similar to kelp and Fucus in both years (Table 2 and Table 3).

**Table 1 biology-13-00874-t001:** The number (*n*), mean δ13C, and δ15N values ± standard deviation of benthic organisms from the Inner Basin (IB) and Sill stations collected in Blagopoluchiya Bay, the Kara Sea, in 2018, 2020, and 2022.

Phylum		Species	Station	2018			2020			2022		
				*n*	δ^13^C‰	δ^15^N‰	*n*	δ^13^C‰	δ^15^N‰	*n*	δ^13^C‰	δ^15^N‰
Porifera		*Polymastia* sp.	Sill	1	−21.7	10.6						
Echinodermata	Asteroidea	*Ctenodiscus crispatus* (Bruzelius, 1805)	IB							2	−18.3 ± 1.2	13.0 ± 0.2
		*Urasterias lincki* (Müller and Troschel, 1842)	IB	2	−17.9 ± 1.3	17.9 ± 2.3	3	−18.6 ± 0.5	19.0 ± 1.4	2	−17.8 ± 0.5	17.1 ± 0.6
			Sill				1	−18.4	17	1	−18.6	15.1
	Ophiuroidea	*Ophiopleura borealis* Danielssen and Koren, 1877	IB	2	−21.8 ± 0.1	11.0 ± 0.2						
			Sill	4	−20.8 ± 0.2	10.7 ± 0.5						
Cnidaria	Actiniaria	*Hormathia digitata* (O.F. Müller, 1776)	IB	5	−20.5 ± 0.6	13.2 ± 1.2	4	−21.47 ± 0.55	14.3 ± 0.4	4	−21.8 ± 0.2	13.2 ± 0.9
			Sill	5	−17.8 ± 0.9	13.6 ± 0.7	1	−19.0	13.3	1	−21.1	14.3
	Gersemia	*Gersemia* sp.	Sill	10	−20.0 ± 0.4	11.3 ± 0.5	4	−19.6 ± 0.6	11.0 ± 0.5	4	−21.8 ± 0.5	11.1 ± 0.5
Mollusca	Bivalvia	*Astarte crenata* (J. E. Gray, 1824)	Sill	6	−20.9 ± 0.5	11.0 ± 0.2	2	−21.5 ± 0.2	12.7 ± 1.1			
		*Bathyarca glacialis* (Gray, 1824)	Sill	9	−21.8 ± 0.2	10.3 ± 0.4						
		*Ennucula tenuis* (Montagu, 1808)	IB	12	−22.1 ± 0.4	7.0 ± 0.1						
			Sill							2	−22.3 ± 0.6	7.3 ± 0.0
		*Portlandia arctica* (Gray, 1824)	IB							1	−22.6	7.8
		Thyasiridae gen. sp.	IB							2	−22.3 ± 0.6	7.8 ± 1.04
Nematoda		Nematoda	IB							1	−22.9	8.4
			Sill							1	−23.5	9.4
Annelida	Polychaeta	*Aglaophamus malmgreni* (Théel, 1879)	IB	10	−20.8 ± 0.6	12.3 ± 0.9	2	−21.5 ± 0.5	13.4 ± 0.4	3	−20.7 ± 0.1	12.8 ± 1.0
			Sill							1	−21.6	13.2
		*Artacama proboscidea* Malmgren, 1866	Sill							5	−21.9 ± 0.2	10.6 ± 0.2
		*Bylgides* sp.	Sill							5	−22.0 ± 0.2	10.6 ± 0.3
		*Capitella capitata* (Fabricius, 1780)	IB							1	−21.0	6.4
		Cirratulidae gen. sp.	Sill							4	−22.0 ± 0.3	8.9 ± 0.1
		Maldanidae gen. sp.	Sill							1	−21.9	11.5
		*Micronephthys minuta* (Théel, 1879)	Sill							1	−21.3	11.5
		*Nothria hyperborea* (Hansen, 1878)	Sill				5	−21.9 ± 0.3	11.7 ± 0.5	1	−20.7	12.9
		*Scoletoma fragilis* (O.F. Müller, 1776)	Sill							3	−22.7 ± 0.1	10.7 ± 0.1
		*Scoloplos armiger* (Müller, 1776)	IB							2	−21.9 ± 0.2	11.2 ± 0.13
			Sill							1	−22.1	10.5
		Spionidae gen. sp.	Sill							1	−21.8	9.1
		*Terebellides stroemii* Sars, 1835	Sill							2	−21.8 ± 0.5	10.4 ± 0.7
	Sipuncula	*Golfingia margaritacea* (Sars, 1851)	IB				1	−19.6	10.6			
			Sill				1	−21.0	10.5	5	−19.8 ± 0.3	10.4 ± 0.2
Crustacea	Amphipoda	*Acanthostepheia malmgreni* (Goës, 1866)	IB							4	−19.0 ± 0.6	12.5 ± 0.4
	Copepoda	Harpacticoida	IB							1	−22.7	9.3
			Sill							1	−23.0	7.9
	Cumacea	Cumacea	IB							1	−20.2	9.5
			Sill							1	−19.5	8.2
	Tanaidacea	Tanaidacea	Sill							1	−20.4	9.1
	Caridea	*Eualus gaimardii* (H. Milne Edwards, 1837)	Sill				1	−20.7	10.1	2	−21.6 ± 0.3	11.3 ± 1.1
		*Lebbeus polaris* (Sabine, 1824)	IB	6	−19.8 ± 0.4	11.8 ± 0.7				2	−20.8 ± 2.1	9.8 ± 0.1
		*Pandalus borealis* Krøyer, 1838	Sill				1	−20.7	9.7			
		*Sabinea septemcarinata* (Sabine, 1824)	Sill				1	−20.7	14.8			
		*Sclerocrangon boreas* (Phipps, 1774)	Sill							2	−20.7 ± 0.1	13.8 ± 0.2
	Brachyura	*Chionoecetes opilio* (Fabricius, 1788)	IB	20	−20.7 ± 0.5	12.9 ± 0.8	9	−21.2 ± 0.4	12.4 ± 1.3	3	−20.6 ± 0.3	11.7 ± 0.2
			Sill	30	−20.5 ± 0.6	12.3 ± 0.6	26	−21.3 ± 0.3	13.0 ± 1.1	3	−21.2 ± 0.3	12.3 ± 0.1
Chordata	Osteichthyes	*Anisarchus medius* (Reinhardt, 1837)	Sill	2	−21.9 ± 0.2	14.0 ± 0.3				2	−21.9 ± 0.3	12.7 ± 0.2
		*Boreogadus saida* (Lepechin, 1774)	Sill	1	−22.7	12.5						
		*Gymnocanthus tricuspis* (Reinhardt, 1830)	Sill	1	−21.2	14.2						
		*Hippoglossoides platessoides* (Fabricius, 1780)	IB							1	−22.7	11.0
			Sill							1	−21.9	12.9
		*Leptagonus decagonus* (Bloch and Schneider, 1801)	Sill	1	−22.5	16.2				2	−21.3 ± 1.3	13.6 ± 2.5
		*Lycodes pallidus* Collett, 1879	IB	2	−21.3 ± 0.8	14.1 ± 0.2						
			Sill	2	−21.3 ± 0.8	13.7 ± 0.1						
		*Triglops* sp.	Sill	1	−22.6	13.8				1	−22	12

**Table 2 biology-13-00874-t002:** The mean ± standard deviation of δ^13^C and δ^15^N values of macrophyte algae analyzed from different stations in Blagopoluchiya Bay, the Kara Sea, collected in 2020 and 2022.

Year	Station	Algae	*n*	δ^13^C‰	δ^15^N‰
2020	Shore	Fucus	5	−20.2 ± 2.4	4.9 ± 0.4
2020	Sill	Kelp	2	−20.4 ± 2.7	5.2 ± 1
2020	Inner Basin	Kelp	2	−22.4 ± 0.6	4.9 ± 0.5
2022	Sill	Kelp	1	−18.6	4.7
2022	Inner Basin	Kelp	2	−17 ± 2.2	3.7 ± 0.2
2022	Sill	Red algae	1	−34.9	2.4
2022	Inner Basin	Red algae	2	−21.7 ± 2.7	4.4 ± 0.9
2020	Shore	Red algae 1	3	−22.2 ± 1.2	2.4 ± 0.1
2020	Sill	Red algae 2	2	−22.1 ± 2.5	4.2 ± 0.9
2020	Inner Basin	Red algae 3	1	−36.3	4.6
2020	Inner Basin	Red algae 4	1	−37	5.7
2020	Inner Basin	Red algae 5	1	−34.3	6.7
2022		Phytoplankton *	5	−25.3 ± 0.2	4.2 ± 0.1
2020		Phytoplankton *	3	−27 ± 0	5.8 ± 0.3

* Phytoplankton values were calculated by subtracting 1‰ from δ^13^C and 3.4‰ from δ^15^N *Calanus* spp. values (Table 3) for the respective years.

**Table 3 biology-13-00874-t003:** The mean ± standard deviation of δ^13^C and δ^15^N values of a number (*n*) of pelagic animal species analyzed from Blagopoluchiya Bay, the Kara Sea, collected in 2020 and 2022.

Phylum		Species	ID	2020			2022		
				*n*	δ^13^C‰	δ^15^N‰	*n*	δ^13^C‰	δ^15^N‰
Crustacea	Copepoda	*Calanus* gen. sp.	Calanus	3	−26 ± 0.0	9.2 ± 0.3	5	−24.3 ± 0.2	7.6 ± 0.1
	Euphausiacea	Euphausiidae gen. sp.	Euphaus	4	−24.6 ± 0.4	9.9 ± 0.5	5	−23.2 ± 0.2	8.4 ± 0.9
	Mysida	*Mysis* sp.	Mysis	1	−23.1	9.6	4	−22.3 ± 0.4	10.2 ± 0.7
	Brachyura	*Chionoecetes opilio* (Fabricius, 1788) Zoea	ChOplZoe	1	−24.01	7.7	3	−24.1 ± 1.1	7.1 ± 0.3
Chaetognatha		*Parasagitta elegans* (Verrill, 1873)	Chaetogn	2	−25.5 ± 0.0	12.8 ± 0.9	2	−24.2 ± 1.3	10.1 ± 1.0

The TP of consumers did not significantly differ (chi-squared = 4.1, df = 3, *p*-value = 0.2) when it was calculated using phytoplankton values approximated from the respective years or as a bulk from all years. In 2020, the TPs using bulk PP were only slightly higher than the TPs using 2020 PP, and the difference ranged from −0.08 to 0.24 (on average 0.07). In 2022, the TPs using bulk PP were slightly lower than those using 2022 PP, and the difference varied insignificantly, averaging −0.1 (−0.2 to 1). Such differences do not result in different trophic levels of the studied species and, therefore, do not significantly affect the interpretation of the data. The same species sampled in different years did not show substantial differences in their trophic positions and lay within the same trophic levels (Figure 2 and Figure 3, Table 4). All species sampled from both the Sill and the Inner Basin stations had TPs within the same TL. Therefore, the TPs of organisms sampled in 2020 and 2022 were calculated using phytoplankton values from the respective years and their total values for 2018 data, where no Calanus spp. were sampled. The Discussion section of this manuscript presents the reasons for choosing these primary producers for the baseline.

The differences in alphas using different baselines were more profound (chi-squared = 12.7, df = 3, *p*-value = 0.005). In 2020, alphas using bulk PP were, on average, −0.23 (−0.34 to −0.12) lower than when using 2020 PP, although not significantly different (post-hoc *p* = 0.14). However, in 2022, the alphas using bulk PP were significantly different (post-hoc *p* = 0.03), on average, −0.1 (−0.4 to 0) lower than when using 2022 PP. This is related to the different isotopic values of Calanus spp. in these years. Since phytoplankton and Calanus species composition could vary between and within the year, we believe that the bulk sample is a representative baseline for long-living species and should be regarded with care for short-living species. The alpha values of consumers sampled in 2018 should be interpreted with care.

The rescaling model resulted in lower TPs of species below TP4 and higher TPs above it. Rescaled TPs suggest that some species are above TL4 (up to TL 6), and a larger number of consumers had TPs at TL1, which is the primary producers’ level. Indeed, even when using the linear model, some consumers had TPs lower than 2, but fewer and to a lesser extent than when using the rescaled model. Such low TPs can be attributed to errors associated with the selected baseline values, sampling size, and “cleanness” of sampled species. In addition, the rescaled model uses only δ^15^N and does not account for different sources of organic carbon (benthic and planktonic), which can increase TP calculation error and miss the opportunity for additional information about the source. Therefore, all further analyses will be based on two baseline model TPs, using the constant discrimination factor across all TLs.

Pelagic species are not the main focus of this study. However, some were analyzed to approximate the contribution of pelagic sources of organic material in the Blagopoluchiya Bay benthic food web. The trophic positions of sampled pelagic organisms ranged from primary consumers (*Calanus* spp., TP~2) to highly predatory (chaetognaths, TP > 3) (Figure 2). Most planktonic animals have alphas below 0.5, suggesting a preference for phytoplankton sources of organic carbon (Figure 2).

Thirty-six different benthic organisms were analyzed using stable isotope analysis (Table 1), thirteen of which were resampled in different years and twelve from both stations. The trophic positions of benthic organisms (Figure 3, Table 4) span from primary consumers, such as *Capitella capitata* (TP 1.5), to the highest predator *Urasterias lincki* (TP 5.2). The trophic position (TP) and PP source preference did not change for the crabs and most of the benthic species. A sea star, *Urasterias lincki*, remained the highest carnivore in the bay, and the invasive crab remained at a lower trophic level (TP 3.4), showing higher omnivory in its diet.

Most benthic organisms primarily use benthic or a combination of benthic and pelagic-derived organic carbon sources (alpha 0.5 and above, Figure 3). Only a few taxa had alpha values below 0.5, such as meiobenthic organisms Nematoda and Harpacticoida, a few fishes that can feed on pelagic species, and predatory polychaetes that can feed on meiobenthic species.

**Table 4 biology-13-00874-t004:** The trophic position and alpha (in brackets) of benthic organisms from the Inner Basin (IB) and Sill stations collected in 2018, 2020, and 2022. The feeding mode was assigned based on literature data (see Methods and Discussion).

Phylum		Species	ID	2018		2020		2022		Feeding Mode
				Sill	IB	Sill	IB	Sill	IB	
Porifera		*Polymastia* sp.	Polymast	2.7 (0.5)						Suspension feeder
Echinodermata	Asteroidea	*Ctenodiscus crispatus*	CtenodCr						3.4 (1.3)	Deposit feeder
		*Urasterias lincki*	Uraster		4.9 (1)	4.7 (1.2)	5.2 (1.2)	4 (1.2)	4.6 (1.4)	Active predator
	Ophiuroidea	*Ophiopleura borealis*	Ophiuroid	2.8 (0.7)	2.8 (0.5)					Deposit feeder
Cnidaria	Actiniaria	*Hormathia digitata*	Hormath	3.6 (1.1)	3.5 (0.7)	3.5 (1.1)	3.7 (0.8)	3.8 (0.8)	3.6 (0.7)	Suspension feeder
	Gersemia	*Gersemia* sp.	Gersem	2.9 (0.8)		2.9 (1)		2.9 (0.6)		Suspension feeder
Mollusca	Bivalvia	*Astarte crenata*	Astart	2.8 (0.7)		3.3 (0.8)				Suspension feeder
		*Bathyarca glacialis*	Bathyarc	2.6 (0.5)						Suspension feeder
		*Ennucula tenuis*	Ennucula	1.7 (0.5)				1.8 (0.6)		Deposit feeder
		*Portlandia arctica*	Portland						2 (0.5)	Deposit feeder
		Thyasiridae gen. sp.	Thyasirid						2 (0.6)	Deposit feeder
Nematoda	Nematoda	Nematoda	Nematod					2.5 (0.3)	2.2 (0.5)	Deposit feeder
Annelida	Polychaeta	*Aglaophamus malmgreni*	Aglaoph		3.2 (0.6)		3.5 (0.8)	3.5 (0.7)	3.4 (0.9)	Active predator
		*Artacama proboscidea*	ArtacProb					2.8 (0.6)		Deposit feeder
		*Bylgides* sp.	Bylgides					2.8 (0.6)		Active predator
		*Capitella capitata*	CapitCap						1.5 (0.8)	Deposit feeder
		Cirratulidae gen. sp.	Cirratulid					2.3 (0.6)		Deposit feeder
		Maldanidae gen. sp.	Maldanid					3.1 (0.6)		Deposit feeder
		*Micronephthys minuta*	Microneph					3 (0.7)		Active predator
		*Nothria hyperborea*	Nothria			3 (0.7)		3.4 (0.8)		Deposit feeder
		*Scoletoma fragilis*	ScoletFr					2.8 (0.5)		Active predator
		*Scoloplos armiger*	ScolopArm					2.8 (0.6)	3 (0.6)	Deposit feeder
		Spionidae gen. sp.	Spionid					2.4 (0.7)		Deposit feeder
		*Terebellides stroemii*	TerebSt					2.7 (0.7)		Deposit feeder
	Sipuncula	*Golfingia margaritacea*	GolfingMar			2.7 (0.8)	2.7 (1)	2.7 (1)		Deposit feeder
Crustacea	Amphipoda	*Acanthostepheia malmgreni*	AcanthMal						3.2 (1.2)	Active predator
	Copepoda	Harpacticoida	Harpact					2 (0.4)	2.4 (0.5)	Deposit feeder
	Cumacea	Cumacea	Cumacea					2 (1.1)	2.4 (1)	Deposit feeder
	Tanaidacea	Tanaidacea	Tanaidacea					2.3 (0.9)		Deposit feeder
	Caridea	*Eualus gaimardii*	Eualus			2.5 (0.9)		3 (0.7)		Active predator
		*Lebbeus polaris*	Lebbeus		3.1 (0.8)				2.5 (0.8)	Active predator
		*Pandalus borealis*	Pandal			2.4 (0.9)				Active predator
		*Sabinea septemcarinata*	Sabinea			3.9 (0.9)				Active predator
		*Sclerocrangon boreas*	Sclerocr					3.7 (0.9)		Active predator
	Brachyura	*Chionoecetes opilio*	ChOpil	3.2 (0.7)	3.4 (0.7)	3.4 (0.8)	3.2 (0.8)	3.3 (0.8)	3.1 (0.9)	Active predator
Chordata	Osteichthyes	*Anisarchus medius*	Anisarch	3.7 (0.4)				3.4 (0.6)		Active predator
		*Boreogadus saida*	Boreogad	3.3 (0.3)						Active predator
		*Gymnocanthus tricuspis*	Gymnoc	3.8 (0.5)						Active predator
		*Hippoglossoides platessoides*	Hippog					3.4 (0.6)	2.9 (0.5)	Active predator
		*Leptagonus decagonus*	Leptag	4.4 (0.3)				3.6 (0.8)		Active predator
		*Lycodes pallidus*	Lycodes	3.6 (0.5)	3.7 (0.5)					Active predator
		*Triglops* sp.	Triglop	3.6 (0.3)				3.2 (0.6)		Active predator

## 4. Discussion

### 4.1. Primary Producers

The most accepted baseline for trophic calculations using isotopic values is primary consumers at a trophic level of 2 [53]. It is often very hard to find selective herbivores in marine environments and even harder to find the same species in different communities to compare. Therefore, primary producers are often selected [47,76].

The main sources of organic carbon in coastal marine environments include terrestrial and freshwater runoff and different types of marine macro- and microalgae. The harsh Arctic environment in Blagopoluchiya Bay results in minimal and short-lived freshwater and terrestrial vegetation, and its contribution to the benthic food web is negligible since the δ^13^C of terrestrial vegetation is usually near −30‰ [77], which is very far from the carbon isotope values of the studied consumer organisms. Another substantial source of primary production can be microphytobenthos. It was suggested that stable isotope analysis could differentiate between pelagic and benthic microalgae in low latitudes and intertidal zones; however, that is not always the case in high-latitude, deeper benthic ecosystems, where it can have similar values to phytoplankton [78]. In our case, it would be obscured by the enriched in heavy macrophyte algae or by the lighter phytoplankton carbon isotope values. The extent of microphytobenthos’ contribution to the Blagopoluchiya Bay food web is unknown. However, it should be kept in mind that the production of benthic microalgae could be substantial, at least at the shallower Sill station.

Together with ice algae, pelagic primary production (PP) is often considered the main source of organic material in marine environments [79]. Ice algae can be 2–10‰ more enriched in ^13^C than phytoplankton [70,71]. Although ice algae can substantially contribute to primary production in some Arctic communities, our study was conducted in late autumn, when phytoplankton is the most prominent pelagic PP source. It is very hard to sample phytoplankton without suspended dead organic matter, especially near the shore. In addition, the phytoplankton community may vary in different seasons. Selective herbivores such as mussels (*Mytilus* sp.) are considered the best approximation of phytoplankton [80]. In their absence, the closest approximation to selective herbivores was *Calanus* spp. [72] (Table 1 and Table 2).

Carbon from macrophyte algae can enter food webs by being directly grazed when alive or consumed as degraded residues such as deposited detritus and suspended particulate organic matter (POM) [81,82,83]. There is no published data on the composition of species and the abundance of macrophyte algae in Blagopoluchiya Bay. A study from the northern tip of the Novaya Zemlya Archipelago and similar high Arctic bays of the Severnaya Zemlya Archipelago reports a higher proportion of brown algae, followed by red and very few green algae species [84,85]. Throughout our studies in Blagopoluchiya Bay, we mostly found brown algae *Laminaria digitata*, *Saccharina latissima*, and *Fucus distichus* in our trawl samples and as washout debris on the shore. As in other severe Arctic locations, the tidal and top of the subtidal zones are almost lifeless, and most macrophytes are displaced to a zone with depths greater than 3 m [28].

The δ^13^C of macrophytes can vary from −30‰ in some red algae to −3‰ in seagrasses [86]. There are no seagrasses in Blagopoluchiya Bay. Red algae have a broad spectrum of δ^13^C values; some were closer to −30‰ and, therefore, did not contribute to the trophic web, while others had values similar to kelp and Fucus (Table 2). Fucoids are generally considered a less preferable food source than kelp due to toxic or digestibility-reducing compounds [87]. However, there is some evidence that they can have an important role in the food web [88]. The variability in isotope values of kelp and fucoids is considerable and can be attributed to the natural variability of plants and different stages of degradation of sampled detached specimens. These are long-lived algae, and the length of their decomposition on the seafloor is unknown. Therefore, we used the bulk of all sampled specimens in all years as an approximation of macrophyte contribution to the Blagopoluchiya Bay food web.

The alpha values of the animal species sampled in this study support the view that, along with phytoplankton, macrophyte-derived carbon is an important source of organic carbon supply in the cold-water inshore ecosystem with macrophyte algae communities [81,82,83]. For example, in the Antarctic Peninsula benthic fauna, macrophyte production can contribute up to 90% of the PP supply [89]. A substantial contribution of macroalgae to the fjordic food web has been shown, for example, for Arctic Kongsfjorden (Svalbard) [90], Hornsund (Spitsbergen) [91], and Norway’s coastal kelp forest [92]. However, the opposite has been shown for Arctic Kongsfjorden, where the authors suggest a high contribution of microalgae that can be trapped under the kelp canopy [83]. The proportion of pelagic organic carbon deposited on the seafloor and benthic-derived primary production is a challenging and interesting question deserving a separate detailed study. The results presented in this study may contribute to future studies as an example of species feeding on different sources.

### 4.2. Pelagic Organisms

Our separation of benthic and pelagic species is often arbitrary but necessary for sustainable analysis. However, benthic–pelagic coupling is an integral route of energy flow within a system and is a highly discussed subject in the literature [93]. Therefore, our study looks at some pelagic species to highlight the difference between pelagic- and benthic-derived organic carbon in the Blagopoluchiya Bay food web. The most pronounced difference between benthic and pelagic food chains was observed in 2020 (Table 1 and Table 3), where, on average, the δ^13^C values of pelagic species were 4–5‰ lower than those of the benthic. The δ^13^C value differences are smaller in 2022 (Figure 2).

Euphausiids (TP 2.6, alpha 0.4) are described as deposit feeders, filter feeders, and predators, feeding on both benthic and pelagic food items [94]. Mysids (TP 2.6, alpha 0.4) are filter feeders and predators [94]. Due to their diurnal migrations, they use both pelagic and benthic food sources [95]. Therefore, the medium alpha of these organisms supports literature data describing them as both benthic and pelagic feeders. Such a combination of benthic- and pelagic-derived organic material in mostly pelagic species can contribute to increasing the link between the two food webs (benthic–pelagic coupling). In an extensive study of macroalgae contribution to suspended POM (particulate organic matter), a large proportion of macroalgae material was found in the Mysis diet [90].

Chaetognaths (*Parasagitta elegans*), copepods (*Calanus* spp.), and the zoea of the invasive snow crab (*Chionoecetes opilio*) have alpha values close to 0, suggesting a preference for phytoplankton-derived organic carbon. *Calanus* spp. (TP 1.8) feeds on phytoplankton, preferring diatoms and dinoflagellates [96,97]. In the literature, Chaetognatha *Parasagitta elegans* are described as predators [94]. However, some non-carnivorous feeding has previously been suggested [98]. Chaetognaths sampled in 2020 had a TP (3.3) at the secondary consumer trophic level, whereas in 2022, it was lower at the primary consumer TL (TP 2.6). Therefore, at least some phytoplankton may contribute to their diet.

The zoea of *Chionoecetes opilio* (TP 1.9, alpha 0.2) can feed on phyto- and zooplankton, but data on their diet in the wild are limited. Under experimental conditions, they are mostly fed on zooplankton [99,100]. The larvae of another closely related species, *Chionoecetes bairdi*, can feed on phytoplankton [101]. It is hard to say whether the invasive crab larvae impact the local plankton community, but they are new consumers that can prey on and compete for food with other zooplankton.

### 4.3. Benthic Food Web

The biggest variety of organisms was sampled in 2022 due to the sampling effort. However, a few species were sampled in different years and at different stations that can be compared. Most benthic organisms did not show differences in trophic position or organic source preferences between the years and sampling stations (Figure 3, Table 4). Most species’ trophic positions are in accord with their diets described in the literature. The few exceptions are further discussed in detail.

#### 4.3.1. Suspension Feeders

Most suspension feeders were collected from the Sill station, where inorganic matter sedimentation is lower than in the Inner Basin. All suspension feeders in Blagopoluchiya Bay had relatively high trophic positions, close to or above 3. There are no selective herbivorous suspension feeders in the benthic community of the bay.

In 2018, sea sponge *Polymastia* sp. had a TP of 2.7, suggesting suspension feeding on algae and animal items. Soft corals *Gersemia* sp. can be both deposit and filter feeders; detritus, diatoms, and foraminifers can be found in their gastrointestinal tract [102,103]. Their trophic position is also quite high and stable in all years (2.9). The trophic position of suspension-feeding bivalve *Astarte crenata* was high in both 2018 and 2020 at the Sill station (2.8 and 3.3, respectively). *Bathyarca glacialis* had a slightly lower TP of 2.6 in 2018. In the literature, *Bathyarca glacialis* is reported to have a higher proportion of microalgae in its diet [104], while the selectiveness of *Astarte* is unknown [105].

Actiniaria *Hormathia digitata* is a highly carnivorous organism that catches its prey mainly from the water column [106,107]. Strictly speaking, it cannot be considered a truly sessile suspension feeder. It is often found “riding” on top of predatory gastropods and can position its tentacles close to the oral region of the host to uptake food [108]. The feeding activity of gastropods makes more food available to sea anemones living as epibionts, increasing the proportion of benthic organisms available to *H. digitata* [109]. Fatty acid analysis of *H. digitata* in the Barents Sea suggests its diet is based on phyto- and zooplankton, but mainly focused on *Calanus* spp. [110]. In Blagopoluchiya Bay, *H. digitata* had a high trophic position at the secondary consumer level in all years, ranging from 3.5 to 3.8. Its high alpha values (0.7 to 1.1) suggest predominantly benthic-derived organic carbon. It is unlikely that in Blagopoluchiya Bay, *Calanus* spp. constitutes a high proportion of the *H. digitata* diet.

Similarly, all suspension-feeding organisms in Blagopoluchiya Bay had high alpha values, suggesting a higher proportion of benthic-derived organic material (Figure 3, Table 4). Only *Bathyarca glacialis* in 2018 and *Gersemia* sp. in 2022 had medium (close to 0.5) alpha values, reflecting some pelagic-derived organic carbon in their diet at the Sill station. A few studies have shown that in Arctic ecosystems, macrophytes contribute to half or less of the organic material source in the diet of suspension/filter feeders [111,112,113,114]. However, most of these studies have been in more productive seas, such as the Beaufort and Chukchi Seas, with high levels of ice algae blooms, high phytoplankton productivity, and terrestrial organic matter input. Possibly, in the severe ice conditions of Blagopoluchiya Bay, with low phytoplankton productivity and minimal terrestrial organic matter input, suspension feeders rely on suspended macroalgae in the form of particulate organic matter.

#### 4.3.2. Deposit Feeders

Deposit feeders span across several trophic levels, from primary herbivorous consumers (TP 1.5) to high-level carnivores (TP 3.4) (Figure 3, Table 4). Deposit-feeding organisms that were sampled in different years and at different stations did not show significant shifts in their trophic positions.

Polychaetes constitute the majority of species diversity within this feeding mode. *Capitella capitata* (TP 1.5), which is considered to be a non-selective deposit feeder, but its digestive system almost always contains plant fragments, had the lowest TP [115,116]. Similarly low TP is observed for non-selective deposit-feeding polychaetes from the family Cirratulidae (TP 2.3) [115], and deposit- and suspension-feeders from the family Spionidae gen. sp. (TP 2.4) [117]. These polychaetes have trophic positions that suggest a predominantly plant-based diet.

Closer to secondary consumers are polychaete species *Terebellides stroemii* (TP 2.7), *Artacama proboscidea* (TP 2.8), *Scoloplos armiger* (TP 2.8 to 3.0), and polychaetes from the family Maldanidae (TP 3.1). They are reported to typically feed on detritus, usually diatoms or other unicellular algae, as well as small invertebrates, including larvae, as part of the detritus [115,118,119,120].

The highest TP among polychaetes was observed in *Nothria hyperborea*, which was sampled from the Sill station in 2020 and 2022. In both years, it had high TPs (3.0 and 3.4, respectively) at a secondary consumer trophic level. There is little literature information about this species except that it is a deposit feeder [121,122]. However, other representatives of the Onuphidae family can be classified as scavengers with a fairly diverse diet [115].

A different deposit-feeding Annelida, Sipuncula *Golfingia margaritacea* [121,123], was sampled from both stations in 2020 and from the Sill station in 2022. In all cases, it had the same high trophic position of 2.7, closer to the secondary consumer level.

All deposit-feeding bivalves in Blagopoluchiya Bay had low TP, close to 2, at the primary consumer trophic level. The TP of the bivalve *Portlandia arctica* (2.0) aligns with literature describing it as a deposit feeder that consumes microorganisms from plankton or benthos, which are absorbed together with sediments [124,125]. Another bivalve, *Ennucula tenuis*, is described as both a deposit and suspension feeder, primarily feeds on diatoms settling from the water column, as well as flagellates and detritus [59,116,118,126]. Its trophic position was at the level of primary consumers in both 2018 (TP 1.7) and 2022 (TP 1.8) from the Inner Basin and Sill stations, respectively. The feeding habits of *Thyasiridae* gen. sp. (TP 2.0) are diverse, but it is often described as both a deposit and suspension feeder [127,128]. It is hard to determine which feeding mode predominated in the sampled specimens. However, it was found at the Inner Basin station, where suspension feeding is uncommon due to high mineral sedimentation. The low TP of Cumacea *Eudorella emarginata* (TP 2.0 to 2.4) aligns with literature suggesting that the genus Eudorella feeds on phytodetritus and diatoms [129,130]. Another crustacean, Tanaidacea gen. sp. (TP 2.3), is a deposit and suspension feeder, scavenger, and predator [94]. Its low TP in Blagopoluchiya Bay suggests that it is unlikely to be highly predatory but rather, like most species of Tanaidacea, feeds on small invertebrates, detritus, and algae [131].

In literature, Asteroidea *Ctenodiscus crispatus* (TP 3.4) is described as a non-selective feeder on sediment particles, with bacteria and diatoms playing a major role in its diet [132]. However, our study suggests its high trophic position at a secondary consumer level. It probably utilizes a high proportion of animal organisms in its diet.

The most noticeable change in Blagopoluchiya Bay since the invasion of the snow crab is the disappearance of previously dominant brittle stars, Ophiuroidea. However, in 2018, *Ophiopleura borealis* was still present and sampled at both stations, where its trophic position was the same (2.8). *Ophiopleura borealis* is a mobile deposit feeder and opportunistic scavenger [70], which is consistent with the TP above 2, as derived from our data.

Although collected only in 2022, meiobenthic organisms are very interesting due to their low trophic positions (2 to 2.5) and alpha values (0.3 to 0.4), suggesting a higher proportion of microalgae in their diet. Meiobenthic species are the only benthic organisms with such low alpha values, differentiating them from the rest of the benthic fauna. The two general groups of meiobenthic organisms analyzed in our study feed on both benthic and deposited pelagic microalgae, as well as fungi, bacteria, and other microorganisms. *Harpacticoida* (TP 2.0 and 2.4 from the Sill and Inner Basin stations, respectively) have a broad spectrum of feeding habits. Different species feed on a variety of food items such as microalgae, cyanobacteria, flagellates, ciliates, mucoid substances, fungi, and heterotrophic bacteria [133]. Different species of Nematoda (TP 2.5 at both stations) can feed not only on microalgae but also on bacteria, detritus, protozoans, and metazoan prey [134,135]. If microphytobenthos contribution in Blagopoluchiya Bay is substantial, it is unlikely that meiobenthic organisms would selectively exclude it from their diet. Therefore, such low alpha values suggest that meiobenthic organisms feed on microorganisms that do not derive their organic carbon from macrophytes, and the microphytobenthos in Blagopoluchiya Bay is unlikely to be enriched in heavy carbon isotopes.

Most deposit feeders in Blagopoluchiya Bay had high alpha values (0.7 to 1.3), suggesting the predominance of macrophyte-derived organic carbon in their diet. Lower alpha values (0.5–0.6) of some polychaetes and bivalves could be due to their unselective feeding, with a substantial contribution of either deposited pelagic or benthic microalgae. This is especially true for low trophic level bivalves. Some polychaetes with lower alpha values but higher trophic levels (such as *A. proboscidea*, *S. armiger*, and Maldanidae) can ingest meiobenthic organisms as well as microalgae.

#### 4.3.3. Active Predators

The highest predator in Blagopoluchiya Bay is Asteroidea *Urasterias lincki* (Figure 3, Table 4). This species was sampled in all years and from both stations and consistently showed the highest trophic position, ranging from 4.0 to 5.2. There is no data on the diet of *U. lincki*, but based on its relative, *Asterias forbesi*, it likely feeds on mollusks, small crustaceans, worms, and dead fish [97]. Using an underwater towed video platform, this predator was observed feeding on dead snow crabs in Blagopoluchiya Bay.

All fishes in Blagopoluchiya Bay are slightly lower than *U. lincki* but are still highly carnivorous secondary consumers. It has been shown that the Kara Sea *Leptagonus decagonus* (TP 3.6 to 4.4) feeds on benthic and planktonic species [136,137]. *Anisarchus medius*, described as a predatory fish that feeds on small invertebrates, had similarly high TPs in 2018 (3.7) and 2022 (3.4). The same was observed for *Triglops* sp., which is reported to prey on crustaceans, polychaetes, and fish [138], with its TP in 2018 (3.6) and 2022 (3.2) at the secondary consumer level. The food spectrum of *Gymnocanthus tricuspis* (TP 3.8), *Lycodes pallidus* (TP 3.7), and *Hippoglossoides platessoides* (TP 3.4) includes zooplankton, zoobenthos, and nekton [136,138]. The variations observed in fish species cannot be definitively attributed to changes in their diet due to the small sample size. These differences may be due to individual variation. These fishes may prey on small, settled snow crabs, but once crabs grow larger than approximately 20 mm in carapace width, they are not attainable to these fishes.

Active predatory polychaetes in Blagopoluchiya Bay have high trophic positions consistent with secondary consumer trophic levels. There is no detailed literature on *Aglaophamus malmgreni* and *Micronephthys minuta* (TP 3.0), except that they are predatory species [126,139]. However, the Nephtyidae family as a whole is characterized by feeding on small invertebrates, including mollusks, crustaceans, and other polychaetes [115]. *Aglaophamus malmgreni* was sampled in all years from the Inner Basin station and in 2022 from the Sill station, and their TPs (3.2 to 3.5) were very similar. Representatives of the genus *Bylgides* (TP 2.8) are described as predators and detritivores, which could lower their TP [94,115]. *Scoletoma fragilis* (TP 2.8) is a predator and scavenger [139]; however, its TP suggests that it might also ingest some detritus.

Another highly carnivorous species is Amphipoda, *Acanthostepheia malmgreni* (TP 3.2) [140]. A study using fatty acid analysis suggested that the food source for *A. malmgreni* is phytoplankton [141]. However, in our study, the high TP and alpha (1.2) suggest an insignificant contribution of phytoplankton-derived organic carbon in its diet.

All sampled shrimp species are considered predators and scavengers, and some proportion of detritus is present in their diet. *Sabinea septemcarinata* (TP 3.9) feeds on infauna [142]. The stomachs of *Sclerocrangon boreas* (TP 3.7) can contain a large number of polychaetes, mollusks, crustaceans, and brittle stars, as well as algae [140,143]. *Pandalus borealis* (TP 2.4) feeds on detritus, dead, and live animals, including benthic and planktonic organisms [144]. *Eualus gaimardii* can eat detritus, algae, mollusks, copepods, and amphipods [122,143,145]. *Lebbeus polaris* can ingest hydroids, amphipods, and polychaetes [143,145]. We cannot draw any conclusions from the differences in TPs of *E. gaimardii* (2.5 and 3.0 in 2020 and 2022, respectively) and *L. polaris* (3.1 in 2018 to 2.5 in 2022) due to the small sample size, as this can be due to individual variation.

The food preference of the invasive *Chionoecetes opilio* is well-studied in both native and invasive regions. It is considered a scavenger and predator [42,103]. According to stomach analysis, their diet usually consists of various groups of invertebrates, mollusks, and crustaceans, as well as their own species, polychaetes, brittle stars, and algae [40,41,42,43]. A few studies in new habitats note the role of detritus in the snow crab diet [146]. Stomach content analysis of snow crabs from Blagopoluchiya Bay in 2018 revealed the dominance of brittle stars, polychaetes, and detritus in their diet [147]. Snow crabs probably feed on the most numerous, and therefore accessible, species of macrozoobenthos. A large amount of detritus may be a feature of their diet in the invasive habitat [148]. A study by the authors of this paper has shown that by 2022, the brittle stars had disappeared from their stomachs, while bivalves became more prominent. The proportion of detritus remained high, while their trophic (isotopic niche) size did not significantly change between 2018 and 2023 [149]. In our study, its trophic position did not change between 2018 and 2022 at both stations (TP 3.1 to 3.4). However, the availability of prey items has changed in these years [16]. Due to these changes, the shift in its diet must have occurred within the same trophic levels of its prey items. For example, the previously highly available ophiuroids were substituted by polychaetes and bivalves from the same trophic level.

Most of the animals’ trophic positions are in accordance with information from the literature about their feeding preferences. Species sampled in different years and at different stations had similar TPs, with the exceptions of a few fish and shrimp with small sample sizes. These are highly mobile predators that can travel substantial distances, prey on different organisms depending on their availability, and have high individual variability in their diet.

Predatory species mostly derive organic carbon from macrophyte sources (alpha 0.7 to 1.4). This is especially true for starfish, most polychaetes, shrimp, and snow crabs. Slightly lower alpha values (0.5–0.6) were observed for polychaetes *S. fragilis* and *Bylgides* sp., which can ingest detritus and, with it, microalgae and meiobenthic species with lower alpha values. Varying alpha values of fishes (0.3 to 0.8) can be a reflection of their ability to feed on both benthic and pelagic organisms.

### 4.4. Trophic Changes in Blagopoluchiya Bay

The benthic community of Blagopoluchiya Bay was studied at the early stages of the snow crab invasion (2014–2016) and in the following years [16]. Although the crabs were numerous in the early stages of the invasion, they were very small (<20 mm carapace width) and did not prey on larger macro and megabenthic species. In these early years (2014–2016), the benthic community of Blagopoluchiya Bay was similar to that before the invasion. The most noticeable changes occurred in the diversity and biomass of megabenthic organisms after 2018, when the crabs reached a large enough size to effectively feed on larger prey [16]. By 2020, they had grown even further (CW around 40 mm) and evaded the Sigsbee trawl, decreasing their proportional biomass in the sample. However, the decrease in their abundance was also evident from the video data, although less pronounced between 2020 and 2022, after a sharp decrease in 2018. It is interesting to compare how the food web structure altered as the composition and biomass of the most common megabenthic species changed during the invasion.

The average trophic position calculated in this study (in all years and stations) has been used for each megabenthic species considered most common in Blagopoluchiya Bay from 2014 to 2022 [16]. Their average weighted TP contribution was calculated for each year based on their percentage biomass. The weighted average trophic position of each feeding mode guild was calculated using the sum percentage biomass of all species within each feeding mode guild (Table 5). Most of the species reported as most common in trawl samples in Udalov et al. [16] were analyzed in this study, and their respective average TPs were used. However, two species found at the Inner Basin station were not sampled for isotope analysis: active predator *Saduria sabini* was only present in small quantities (0.6% of trawl biomass) in 2004, and predatory gastropod *Colus sabini* was also present in small quantities (up to 6.4% biomass) but in most of the years. In comparison to other active predators like snow crabs with biomass up to 46% and *Urasterias lincki* with biomass up to 63%, their contribution to the total biomass of predators is negligible, and they were excluded from the calculations.

There was a substantial increase in the proportion of active predators at both stations after the invasion of snow crabs. The only large deposit feeders, ophiuroids, have disappeared. Suspension feeders have remained in the same proportion at the Sill station; however, their species composition changed from nutritional bivalves to sea sponges, less desirable food sources for the crabs [16]. At the Inner Basin station, the proportion of suspension feeders has greatly increased, mainly due to the rise of tough-to-ingest sea anemone *Hormathia digitata* and the decrease of other suspension and deposit feeders.

Overall, the average weighted trophic position of all megabenthic species at the Sill station increased by one trophic level (from the high-end second level, where some herbivory existed, to the third level, where mostly secondary consumers remained) and stayed within the same (third) trophic level at the Inner Basin station.

The three feeding mode guilds in this study differentiate organisms based on the method of acquiring food and where the food is found (water column, sediments, or active hunting). There are species from different taxonomic groups and diverse functional traits. Such diversity is reflected in the trophic position range of species within each of these groups: one trophic level in suspension feeders (TP from 2.6 to 3.8), two trophic levels in deposit feeders (TP from 1.5 to 3.4), and three trophic levels within active predators (2.4 to 5.2). Such diversity within feeding mode guilds and trophic overlap between them has been shown for numerous Arctic food webs [60,65,103]. Although such division can shed light on the primary sources of organic material in different habitats, it does not reflect direct competition for food within these guilds.

For example, polychaetes and bivalves represent the most numerous and diverse groups of deposit feeders. Their coexistence could lead to competition for resources, involving not only food but also space. Although most deposit-feeding bivalves have a low trophic position compared to polychaetes, some species occupy similar levels. To study their resource partitioning, a detailed study of each species is needed, including their life histories and habitat preferences in addition to detailed feeding habits studies. However, in the low-productivity conditions of the Kara Sea, and Blagopoluchiya Bay in particular, the density of organisms on the sea floor is quite low. After changes in the species composition and biomass of macrobenthic organisms, there was an increase in meiobenthic organisms [30]. This could be the result of lower predatory pressure and higher food availability, especially after the disappearance of large deposit-feeding brittle stars. Resource partitioning that existed prior to the crabs’ invasion has been disrupted, and the system is undergoing community restructuring that could result in a newly balanced community with different dominant species and feeding habit guilds.

## 5. Conclusions

The main changes in the megabenthos of Blagopoluchiya Bay occurred after 2018, as the crabs grew larger and could feed on mollusks and brittle stars. The invasion of a large predatory snow crab into a previously undisturbed Arctic bay has not affected the trophic position of organisms but has resulted in significant changes in the bay’s food web. There was an overall decrease in the proportion of deposit and suspension feeders. The remaining deposit and suspension feeders are less available as food items for the snow crab, such as the deeply burrowing Sipuncula *Golfingia margaritacea* and the tough-to-chew, low-nutrition *Gersemia fruticosa*, *Hormathia digitata*, and *Polymastia grimaldii*. With the addition of the snow crab, active predators became one of the most abundant feeding mode groups in the Blagopoluchiya Bay food web.

## Figures and Tables

**Figure 1 biology-13-00874-f001:**
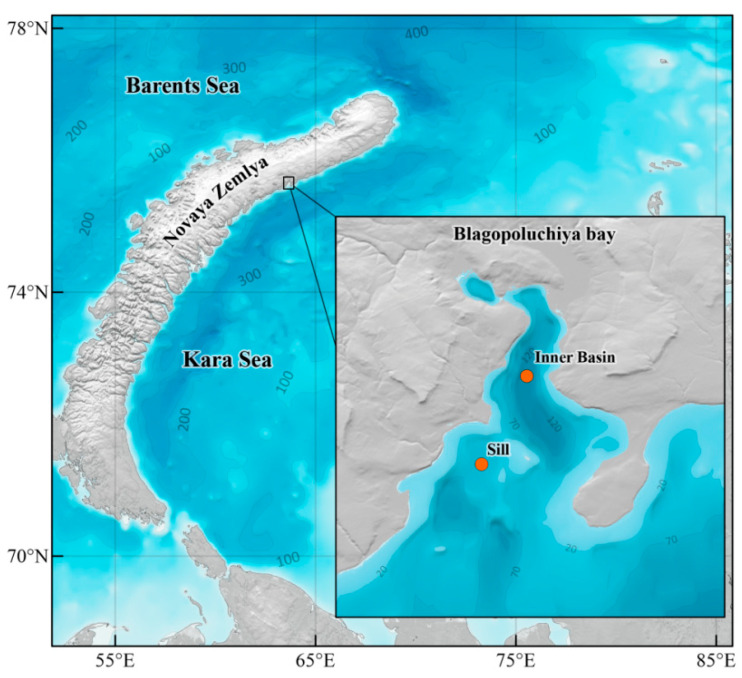
Map of Blagopoluchiya Bay with sampling sites marked by circles.

**Figure 2 biology-13-00874-f002:**
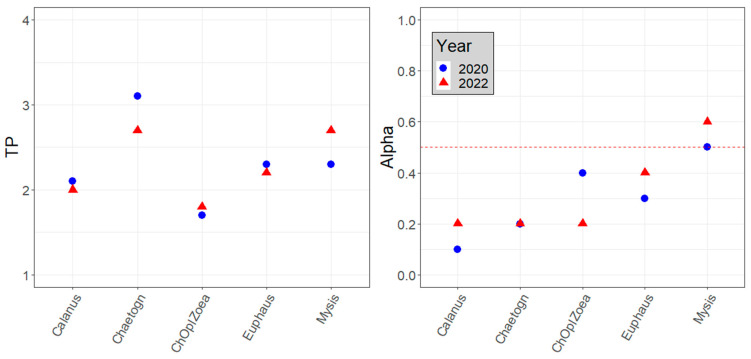
The trophic position and alpha of pelagic animal species from Blagopoluchiya Bay, the Kara Sea, collected in 2020 (blue circles) and 2022 (red triangles). The dashed red line drawn at alpha = 0.5 highlights the preference for macrophyte carbon source (above the line) or phytoplankton carbon source (below the line).

**Figure 3 biology-13-00874-f003:**
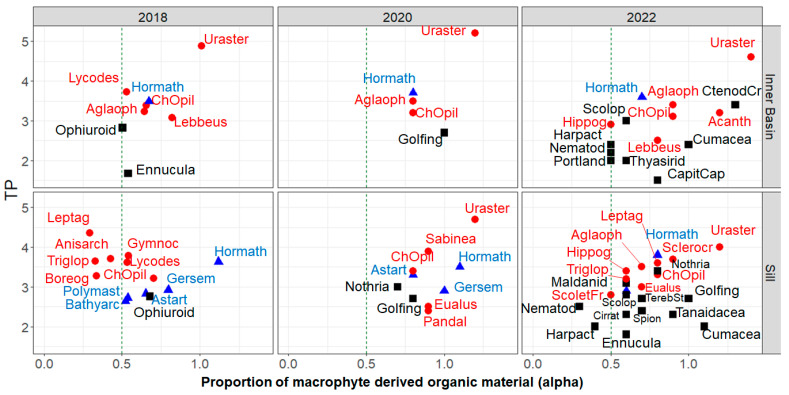
The trophic position and alpha (proportion of macrophyte-derived organic carbon) of benthic animal species from the Inner Basin and Sill stations in Blagopoluchiya Bay, the Kara Sea, collected in 2018 (black squares), 2020 (blue circles), and 2022 (red triangles). The dashed red line drawn at alpha = 0.5 highlights the preference for macrophyte carbon source (above the line) or phytoplankton carbon source (below the line).

**Table 5 biology-13-00874-t005:** The total percentage biomass of the most common organisms caught in trawl samples of Blagopoluchiya Bay (from Udalov et al. [16]), belonging to the three feeding mode guilds, and their respective weighted average trophic positions (*AvWeightTP*).

	**Feeding Mode**	**Value**	**2016**	**2018**	**2020**	**2022**
Sill Station	Deposit feeders	% Biomass	33	1.9	0.3	0.9
		AvWeightTP	2.8	2.7	1.7	1.7
	Suspension	% Biomass	41	5.7	42.9	38.5
	feeders	AvWeightTP	2.7	2.8	2.9	2.8
	Active predators	% Biomass	5.5	86.4	52.7	52.6
		AvWeightTP	3.3	3.3	3.7	3.9
	Total	AvWeightTP	2.8	3.3	3.4	3.4
	**Basin**	**Value**	**2014**	**2018**	**2020**	**2022**
Inner Basin Station	Deposit feeders	% Biomass	67.5	49.1	1	0.1
		AvWeightTP	2.8	2.8	2.6	1.7
	Suspension	% Biomass	1.7	2.3	0	55.3
	feeders	AvWeightTP	3.6	3.6	0	3.6
	Active predators	% Biomass	17.6	46,6	97.4	36.7
		AvWeightTP	4.7	3.1	4.2	4.0
	Total	AvWeightTP	3.2	3.1	4.2	3.7

## Data Availability

Raw data of stable isotope analysis is provided in the Appendix A with this article.

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
