# Peer review of "Trophic Position Stability of Benthic Organisms in a Changing Food Web of an Arctic Fjord Under the Pressure of an Invasive Predatory Snow Crab, Chionoecetes opilio"

_biology, 2024, doi:10.3390/biology13110874_

Round 1

Reviewer 1 Report

Comments and Suggestions for Authors

Review of the manuscript 'Trophic position stability of benthic organisms in a changing food web of an Arctic fjord under the pressure of an invasive predatory snow crab, Chionoecetes opilio' by Anna K. Zalota, Polina Yu. Dgebuadze, Alexander D. Kiselev, Margarita V. Chikina, Alexey A. Udalov , Daria V. Kondar, Alexey V. Mishin, Sergey M. Tsurikov

The authors of this research paper conducted an analysis of the ecological impact of the invasive snow crab, Chionoecetes opilio, on the Kara Sea ecosystem, specifically in Blagopoluchiya Bay. This study provides a unique opportunity to observe the effects of an invasive species in an environment relatively free from other anthropogenic stressors. By employing stable isotope analysis, the authors examined the trophic interactions among the region's benthic species at various stages of the crab's invasion. The authors found that significant alterations occurred within the trophic web, particularly among megabenthic species. The previously dominant deposit-feeding ophiuroids experienced a notable decline in abundance, indicating that the snow crab's introduction may have severely disrupted the existing ecological balance. Despite this dramatic change, the overall trophic position of the remaining benthic species and their preferred sources of primary production remained relatively constant, suggesting that certain ecological roles were resilient to the invasion. Furthermore, the study revealed that Urasterias lincki maintained its status as the highest trophic-level carnivore in the bay. The diets of the consumers that persisted during the invasion reflected changes primarily within their own trophic levels, without significant shifts into higher or lower categories.

The results of this study may have important implications for managing invasive species and understanding predator-prey dynamics in marine ecosystems. This finding underscores the potential for invasive species to alter not just the population dynamics but also the structural complexity of marine food webs. The shift in the food web dynamics may suggest long-term ecological consequences, affecting biodiversity and ecosystem function. Furthermore, the research points to the necessity for continued monitoring of the Kara Sea to investigate the long-term ecological impacts of the snow crab invasion, including potential cascading effects on lower trophic level organisms and overall ecosystem stability.

The paper is well written and illustrated and I have only a few recommendations to improve the text.

Introduction

P 2 L 47: The authors should include examples of the negative effects of invasive species on native biodiversity, ecosystem services, and specific case studies that illustrate these impacts.

P 2 L 50: The authors should provide specific examples of invasive species that have led to new commercial opportunities or tourism growth, along with evidence supporting these claims.

P 2 L 61-63: The authors should specify how a slowdown in sea ice formation in autumn and its earlier destruction in spring-early summer facilitated the introduction of the snow crab. What is the main pathway of this invasion? Please, provide relevant citations.

P 2 L 64-66: The authors mention about the invasion of C. opilio in the Barents Sea. They should provide information on the impact of this species on the local ecosystem and biological resources. I have checked the Google scholar and found a paper devoted to the related topic (Dvoretsky A.G., Dvoretsky V.G. 2015. Reviews in Fish Biology and Fisheries 25, 297–322).

P 2 L 84-85: What specific environmental factors were ruled out as causes for the changes observed in the benthic community? The authors should provide more details on the environmental parameters that were monitored and explain why they were determined not to contribute to the observed changes.

Methods

P 4 L 155-157: The authors should clarify the criteria used to select the two monitoring stations for sample collection in terms of biodiversity or environmental conditions to provide context for why these stations are significant for the study.

P 4: The authors should include descriptions of sediment characteristics (grain size, organic content) at both sites as these characteristics might influence the composition and abundance of benthic fauna.

Results

P 6 L 257-258: Given that the macrophyte algae were not identified at the species level, how might this lack of specificity affect the accuracy of the trophic position calculations? This should be explained in the Materials and Methods.

P 7 Table 1: It would be interesting to know if the differences between years were significant or not.

P 10 Table 2: The authors mention the location "shore", but it was not described in the materials and methods. It would be useful to include this location in the map of the study area.

P 10 L 271-272: In the methods, the authors should provide a detailed description of the method used to calculate the phytoplankton data with relevant citations (L 276-280).

P 10 L 299-301: What statistical analyses were applied to assess the significance of differences in trophic positions calculated through various baseline approaches?

Discussion

P 16 L 414-418: How do the findings on the contribution of macrophyte-derived carbon to the inshore ecosystem in Blagopoluchiya Bay compare to those observed in similar Arctic or sub-Arctic ecosystems? The authors should compare their data with other locations.

P 16 L 428-430: Given that Euphausiids and Mysids are categorized as deposit feeders and predators but also utilize both food sources, the authors should discuss how their feeding behavior might affect the trophic structure in Blagopoluchiya Bay.

P 16 L 438: It would be interesting to discuss how the change in trophic position of Chaetognaths from 2020 to 2022 reflect changes in food web dynamics and shifts in nutrient cycling in Blagopoluchiya Bay.

P 17 L 481. The authors should compare their data on throphic position of suspension feeders with other Arctic or sub-Arctic systems.

P 18 L 522. Considering that Polychaetes and bivalves represent distinct groups within deposit feeders, it would be useful to discuss resource partitioning between these groups to determine how their feeding strategies may co-exist and the potential impacts on community structure.

P 21 L 688. The authors should discuss the ecological impacts of the disappearance of large deposit feeders, like ophiuroids, on nutrient dynamics and energy flow within the benthic ecosystem.

Author Response

Dear reviewer,

thank you very much for taking the time to read our manuscript, for your kind assessment and very helpful remarks. We have tried to comply with them to the maximum. Following are our individual replies to the comments. 

Introduction

P 2 L 47: The authors should include examples of the negative effects of invasive species on native biodiversity, ecosystem services, and specific case studies that illustrate these impacts.

Reply: We have added “The spectrum of negative impacts of invasions on ecosystems can be very diverse. Invasions can lead to the displacement of native species due to changes in the structure and functioning of native ecosystems, they can be environmental engineers, altering physical properties of the habitat and bring along with them parasites and pathogens [Petrosyan et al., 2018; Rilov, Crooks, 2009; Wallentinus and Nyberg, 2007, Gaevskaya, 2004].”

P 2 L 50: The authors should provide specific examples of invasive species that have led to new commercial opportunities or tourism growth, along with evidence supporting these claims.

Reply: added an example of the Red King crab in the Barents Sea “Marine invasive species can positively impact ecosystem services by establishing new commercial fishing, recreation, and tourism services, such as the King crab in the Barents Sea, that has created a large commercial fishing and turism industry both in Russia and Norway [Jorgensen, Spiridonov (2013,), (Lorentzen et al., 2018), 5,8].”

P 2 L 61-63: The authors should specify how a slowdown in sea ice formation in autumn and its earlier destruction in spring-early summer facilitated the introduction of the snow crab. What is the main pathway of this invasion? Please, provide relevant citations.

Reply: This has facilitated the introduction of an invasive crab by giving enough time and suitable conditions for the crab larvae, that inflows from the Barents Sea and recently hatches in the Kara Sea, to develop and settle [Lipukhin 2024].

P 2 L 64-66: The authors mention about the invasion of C. opilio in the Barents Sea. They should provide information on the impact of this species on the local ecosystem and biological resources. I have checked the Google scholar and found a paper devoted to the related topic (Dvoretsky A.G., Dvoretsky V.G. 2015. Reviews in Fish Biology and Fisheries 25, 297–322).

Reply: we have added “Up to date there is no direct evidence of its negative impact on the Barents Sea benthic communities, however such large predator is expected to have adverse effects especially with further growth and expansion of its population [(Zakharov et al., 2024), (Kaiser et al., 2018)]”

P 2 L 84-85: What specific environmental factors were ruled out as causes for the changes observed in the benthic community? The authors should provide more details on the environmental parameters that were monitored and explain why they were determined not to contribute to the observed changes.

 Reply: The abiotic parameters and how they could have affected the changes in the benthic communities in Blagopoluchiya Bay were presented and discussed in our earlier work (Udalov, A.A.; Anisimov, I.M.; Basin, A.B.; Borisenko, G.V.; Galkin, S.V.; Syomin, V.L.; Shchuka, S.A.; Simakov, M.I.; Zalota A.K.; Chikina, M.V. Changes in benthic communities in Blagopoluchiya Bay (Novaya Zemlya, Kara Sea): The influence of the snow crab. Biological Invasions 2024. https://doi.org/10.1007/s10530-024-03388-1), where the main conclusion of the study was that the crab is responsible for the observed changes. Therefore in this manuscript we assume that the changes in the benthic communities are due to crab’s invasion, and regard it as a given. In this work we tried to look at one of the aspects of how these changes have affected the communities, ie in trophic interactions.

We have added a bit more details about how abiotic conditions are not the reason for these drastic changes in benthic fauna in the introduction to set the scene.

“No consistent trends in abiotic parameters, such as ice melting, turbidity, temperature, and nutrients were observed during the observation period in the north-western part of the sea along the coast of the Novaya Zemlya Archipelago and Blagopoluchiya Bay [Udalov et al. 2024]. It has been shown that the changes in benthic communities of the bay cannot be attributed to environmental factors; and that they are the result of the establishment and development of the snow crab population [Udalov et al., 2024].”

Overall throughout the text we tried to highlight that this work is part of a series of papers dedicated to the changes in Blagopoluchiya Bay due to the crab invasion.

Methods

P 4 L 155-157: The authors should clarify the criteria used to select the two monitoring stations for sample collection in terms of biodiversity or environmental conditions to provide context for why these stations are significant for the study.

Reply:We have added “The samples were collected at two monitoring stations that have been sampled numerous times throughout the years: the Inner Basin station, which is located in the deepest inner part of the bay at 150 - 170 m, with high levels of inorganic matter sedimentation; and the shallower SILL station, located near the exit of the bay at a depth of 63 - 72 m (Fig. 1), which has lower levels of sedimentation, and is more connected to the open Kara Sea. The two stations have different benthic communities, with greater proportion of suspension feeders at the Sill station and larger proportion of deposit feeders at the Inner Basin station [Udalov et al., 2024].

P 4: The authors should include descriptions of sediment characteristics (grain size, organic content) at both sites as these characteristics might influence the composition and abundance of benthic fauna.

Reply: This study does not aim to describe the composition and the abundance of benthic fauna. These parameters are used based on our earlier study, which we cite throughout the text.

Udalov, A.A.; Anisimov, I.M.; Basin, A.B.; Borisenko, G.V.; Galkin, S.V.; Syomin, V.L.; Shchuka, S.A.; Simakov, M.I.; Zalota A.K.; Chikina, M.V. Changes in benthic communities in Blagopoluchiya Bay (Novaya Zemlya, Kara Sea): The influence of the snow crab. Biological Invasions 2024. https://doi.org/10.1007/s10530-024-03388-1

This earlier work has all available information on the grain size and composition as well as species composition and abundance. There we also give information and discuss changes in abaiotic conditions during the study period.

However we attempted to highlight this study more in our citations, so that the readers could easily find the needed information.

Results

P 6 L 257-258: Given that the macrophyte algae were not identified at the species level, how might this lack of specificity affect the accuracy of the trophic position calculations? This should be explained in the Materials and Methods.

Reply: We have added in the methods section “Since there are no known species specific primary consumers in the Blagopoluchiya bay benthic fauna, the species composition of macrophytes was not identified. In addition most of the sampled material were fragments of partially decomposed algae, that would be hard to identify.”

P 7 Table 1: It would be interesting to know if the differences between years were significant or not.

Reply:  Indeed this is one of the unfortunate weaknesses of our study. However, due to remoteness of the study area and scarce resources we could not perform multiple sampling (it takes a lot of time to process samples before we can sample the second time, and the expedition would have had to be dedicated solely to this work, which our finance do not allow). In addition a lot of sampled species are rare. These conditions did not allow us to get large enough sample sizes to able to perform parametric statistical analysis. In addition, only a few species are sampled from all years from the same station, so it would be hard to compare results as a bulk. Snow crab is the only species that was sampled at all times, from all stations and in sufficient numbers. We have looked at the changes in the feeding habits of this crab in our different work, that was recently published in this journal, and where parametric and Basian statistic analysis are employed to compare the data.

Kiselev, A.D.; Zalota, A.K. Changes in the Diet of an Invasive Predatory Crab, Chionoecetes opilio, in the Degrading Benthic Community of an Arctic Fjord. Biology 2024, 13, 781. https://doi.org/10.3390/biology13100781

We have attempted to highlight this study more in the discussion of the opilio feeding habits section.

P 10 Table 2: The authors mention the location "shore", but it was not described in the materials and methods. It would be useful to include this location in the map of the study area.

Reply: As it is only two samples and only macrophytes that were collected from the shore, we think that showing it on the map could confuse the readers, and we prefer to keep just the two main stations on the map. However in the methods we added specific location of this shore:

“Samples of macrophyte algae were collected in 2020 and 2022 from trawl samples and the washout piles on the eastern shore opposite the Inner Basin station.”

P 10 L 271-272: In the methods, the authors should provide a detailed description of the method used to calculate the phytoplankton data with relevant citations (L 276-280).

Reply: we have moved the description of the calculations from result to the methods “To describe the baseline (trophic level 1, primary producers (PP)) of the trophic web, we have used the most abundant and large macrophyte species, Kelp and Fucus, as benthic primary producers. In both 2020 and 2022, these species had similar δ13C and δ15N value ranges (Table 2). These are long-living species; therefore, we used values from both years and applied them to all data. For pelagic primary producers, we calculated phytoplankton isotopic values from Calanus spp. (that feed predominantly on phytoplankton [71] by subtracting trophic step discrimination factor of 1‰ from δ13C values and 3.4‰ from δ15N values (Tables 2,3) [47,72].”

P 10 L 299-301: What statistical analyses were applied to assess the significance of differences in trophic positions calculated through various baseline approaches?

 Reply: added “Non-parametric Kruskal-Wallis rank sum test was used to check the differences between TP and alpha values obtained using different baselines, followed by Dunn multiple comparison test with p-values adjusted using the Holm method.” in the methods section. In the results we have added statistical tests results and corrected the average and maximum minimum values that were unfortunately incorrect. However the main results remained the same.

Discussion

P 16 L 414-418: How do the findings on the contribution of macrophyte-derived carbon to the inshore ecosystem in Blagopoluchiya Bay compare to those observed in similar Arctic or sub-Arctic ecosystems? The authors should compare their data with other locations.

Reply:  we have added “The alpha values of the animal species sampled in this study support the view that along with phytoplankton, macrophyte-derived carbon is an important source of organic carbon supply in the cold water inshore ecosystem with macrophyte algae communities [77-79]. For example in Antarctic Peninsula benthic fauna macrophyte production can contribute up to 90% of PP supple [Dunton 2001]. Substantial contribution of macroalgae into fjordic food web has been shown for example for arctic Kongsfjorden (Svalbard) [(Buchholz & Wiencke, 2016)] and Hornsund (Spitsbergen) [SokoÅ‚owski et al. 2014], and Norway coastal kelp forest [(Fredriksen, 2003)]. However, the opposite has been shown for Arctic Kongsfjorden, where the authors suggest high contribution of microalgae that can be trapped under the kelp canopy [(Paar et al., 2019)].”

P 16 L 428-430: Given that Euphausiids and Mysids are categorized as deposit feeders and predators but also utilize both food sources, the authors should discuss how their feeding behavior might affect the trophic structure in Blagopoluchiya Bay.

Reply: Since this study is mostly concerned with feeding habits of benthic organisms, we would like to keep the discussion of pelagic to minimum. However, Thank you for pointing out that these species that feed both on benthic and pelagic derived organic material are a good example of benthic pelagic coupling in pelagic foodweb. We have added

“Such combination of benthic and pelagic derived organic material in mostly pelagic species can contribute to increase the link between the two foodwebs (benthic pelagic coupling). In an extensive study of macroalgae contribution to suspended POM (particulate organic matter), large proportion of macroalgae material was found in the Mysis diet [(Buchholz & Wiencke, 2016)].”

P 16 L 438: It would be interesting to discuss how the change in trophic position of Chaetognaths from 2020 to 2022 reflect changes in food web dynamics and shifts in nutrient cycling in Blagopoluchiya Bay.

Reply: This study is concerned mostly with benthic foodweb, however, a few pelagic species are presented to highlight different sources of organic carbon available to benthic organisms. Indeed the difference in Chaetognaths could be interesting, but there is very little sample size in this study and it would be hard to interpret the data. Indeed the first author is currently involved in a large study of Chaetognaths feeding in the Barents Sea and from that data, it is clear that any conclusions from just a couple of samples are very farfetched. Therefore, we think that any interpretations of pelagic species feeding is out of scope of this manuscript and are not supported by enough data.

P 17 L 481. The authors should compare their data on throphic position of suspension feeders with other Arctic or sub-Arctic systems.

Reply: We have added “A few studies have shown that in the Arctic ecosystems macrophytes contribute to half or less  of organic material source in the diet of suspension/filter feeders [(McMahon et al., 2021), (Bridier et al., 2021); Dunton and others (2012), (Harris et al., 2018)]. However, most of these studies have been in more productive sea, such as Beaufort and Chukchi Seas, with high levels of ice algae blooms, high phytoplankton productivity and terrestrial organic matter input. Possibly in the severe ice conditions of Blagopoluchiya bay, low phytoplankton productivity, and minimal terrestial organic matter input, suspension feeders rely in suspended macroalgae in the form of particulate organic matter.”

P 18 L 522. Considering that Polychaetes and bivalves represent distinct groups within deposit feeders, it would be useful to discuss resource partitioning between these groups to determine how their feeding strategies may co-exist and the potential impacts on community structure.

P 21 L 688. The authors should discuss the ecological impacts of the disappearance of large deposit feeders, like ophiuroids, on nutrient dynamics and energy flow within the benthic ecosystem.

Reply: We have combined the last two comments and written two new paragraphs discussing resources partitioning and brittle star involvement at the end of the discussion section.

“The three feeding mode guilds in this study differentiate organisms based on the method of acquiring food and where the food is found (water column, sediments, or active hunting). There are species from different taxonomic groups and diverse functional traits. Such diversity is reflected in the big trophic position range of species within each of these groups: one trophic level in suspension feeders (TP from 2.6 to 3.8), two trophic levels in deposit feeders (TP from 1.5 to 3.4), and three trophic levels within active predators (2.4 to 5.2). Such diversity within feeding mode guilds and trophic overlap between them has been shown for numerous arctic foodwebs [(McTigue & Dunton, 2014; WÅ‚odarska-Kowalczuk et al., 2019; Zinkann et al., 2021)]. Although such division can shed light on the primary sources of organic material in different habitats, it does not reflect direct competition for food within these guilds.

For example, polychaetes and bivalves represent the most numerous and diverse groups of deposit feeders, their co-existence could lead to competition for resources. Such competition would involve not only food resources but also space. Although, most deposit feeding bivalves have low trophic position in comparison to polychaetes, there are some species that are on similar levels. To study their resources partitioning, detailed study of each species is needed including its life histories, habitat preferences in addition to detailed feeding habits studies. However, in the low productivity conditions of the Kara Sea, and Blagopoluchiya Bay in particular, the density of organisms on the sea floor is quite low, especially after the rapid decline observed since the invasion of the snow crab. After the decline in macrobenthic organisms there was an increase in meiobenthic organisms [(Lepikhina et al., 2022)]. This could be the result of lower predatory pressure and higher food availability, especially after the disappearance of large deposit feeding brittle stars. Resource partitioning that existed prior to the crab’s invasion has been disrupted and the system is undergoing community restructuring that could result in a new balanced community with different dominant species and feeding habit guilds.”

Reviewer 2 Report

Comments and Suggestions for Authors

The manuscript (MS) has indeed serious flaws, additional experiments needed, research not conducted correctly to determine trophic level changes under influence of an invasive crab which was introduced into Barents Sea in 2014, and then occurrence of the crab in study area of Kara Sea in the Arctic region. 

However, the background data before the crab introduction or available data only in 2014 and 2016 were not sufficient for elucidating the aim of the MS,

A complete background of historical data of the snow crab is limited or well not known for the study area,

A food web and food pyramid must be outline including the snow crab,

Benthic community could change with the recent drastic change in the marine environment including abiotic factors, rather than the snow crab alone,

The MS presentation could be considered a report without the trophic indices and statistical analyses,

Benthic community is influenced easily by the anthropogenic factors inducing also hydrographic variables in a year,

Food content of the snow crab is missing for the present study,

Energy flow in math and metabolic rates (excretion, assimilation, absorption rates, etc) are missing for the crab,

For the stability purposed in the MS, a balanced model could need for input and output

These comments above are some of the majority to recommend the MS needing a major revision.

Author Response

Dear reviewer,

thank you very much for taking the time to read our manuscript. We are sorry to learn that you consider it lacking in data, accurate research and to possess serious flaws. We are not certain why specifically our research is not correct to determine trophic levels. We have applied a widely used stable isotope technique and calculations presented by Post (2002) that are also widely used in such studies. However, we feel that many of your more specific comments are very fair and indeed would improve our understanding of the ongoing changes and their consequences in Blagopoluchiya bay. However, unfortunately all that research would not fit into one journal paper, but would result in a large theses. Therefore we have already published papers on some of the issues you raise in you comments (Udalov et al 2024; Kiselev, Zalota 2024) and we are working on future papers dedicated to the last few of your comments. We tried to hilghlight more these published works in out text, and hopefully it will be more apparent to the readers that this paper is one of a series of works dedicated to this issue and the bay. Following are our replies to you specific comments:

Reply: The crab was found in the Barents Sea in 1996 and found in Kara Sea in 2012. Line 64-65 “The snow crab opilio, Chionoecetes opilio, was first found in the Barents Sea in 1996 64 [20]. After rapid population growth and spread, it entered the Kara Sea, where it was 65 found in 2012 [21].” It was first found in Blagopoluchiya Bay in 2014. Line73 “Snow crabs settled in Blagopoluchiya Bay in 2014 [13].”

However, the background data before the crab introduction or available data only in 2014 and 2016 were not sufficient for elucidating the aim of the MS,

Reply: Indeed we present data only starting from 2014 (however earlier data does exist but for different stations of Blagopoluchiya Bay) however first findings of nonindigenous species rare coincide with the beginning of their impact. In 2014 crabs were not numerous and very small (few mm), in 2016 more numerous but still very small to prey on large macro and megabenthic species. The changes occurred after 2018. This has been shown in our previous paper cited throughout our work: Udalov, A.A.; Anisimov, I.M.; Basin, A.B.; Borisenko, G.V.; Galkin, S.V.; Syomin, V.L.; Shchuka, S.A.; Simakov, M.I.; Zalota A.K.; Chikina, M.V. Changes in benthic communities in Blagopoluchiya Bay (Novaya Zemlya, Kara Sea): The influence of the snow crab. Biological Invasions 2024. https://doi.org/10.1007/s10530-024-03388-1

Therefore data from 2014 and 2016 is representative of the situation prior to crab invasion. This has been discussed in the manuscript:

Lines 57-69Until recently, the Arctic 57 Kara Sea was one such place that had low human impact and no non-indigenous species 58 in its benthic ecosystem [13]. In fact, the distribution and species composition of the Kara 59 Sea benthic communities observed in the 20s and 30s of the XXth century remained no-60 tably stable until the 2010s [14-18].”

Lines 73-85 “Snow crabs settled in Blagopoluchiya Bay in 2014 [13]. In 2016, a large number of 73 small (carapace width CW ~15 mm) juvenile crabs were caught in trawl samples, and this 74 single cohort grew throughout the years without any new settlements [13,22,27]. By 2018, 75 crabs grew (CW ~30 mm), and their abundance was more than 20 crabs per 10 m2. In 76 2020, their number crashed (less than ten crabs per 10 m2) and further decreased in 2022 77 [13]. 78

Blagopoluchiya Bay's benthic communities were studied before and during the in-79 vasion [13,28]. The changes in benthic communities occurred at various levels and were 80 first observed after 2018 when the crab reached a large abundance and size that could 81 feed on larger prey such as molluscs and brittle stars. While changes in macrobenthos 82 were relatively weak, changes in megabenthos were dramatic. By 2022, not a single 83 specimen of previously dominant ophiuroids was found in the bay [13]. These changes 84 cannot be attributed to environmental factors; they are the result of the establishment and 85 development of the snow crab population.”

Lines 635-646 “The benthic community of Blagopoluchiya Bay was studied at the early stages of the 635 snow crab invasion (2014-2016) and in the following years [13]. Although the crabs were 636 numerous at the early stages of the invasion, they were very small (<20 mm carapace 637 width) and did not prey on larger macro and megabenthic species. In these early years 638 (2014-2016), the benthic community of Blagopoluchiya Bay was similar to that before the 639 invasion. The most noticeable changes occurred in the diversity and biomass of mega-640 benthic organisms after 2018, when the crabs reached large enough sizes to effectively 641 feed on larger prey [13]. By 2020, they had grown even further (CW around 40 mm) and 642 had evaded the Sigsbee trawl, decreasing their proportional biomass in the sample. 643 However, the decrease in their abundance was also evident from the video data, although less pronounced between 2020 and 2022, after a sharp decrease in 2018. It is in-645 teresting to compare how the food web structure altered as the composition and biomass 646 of the most common megabenthic species changed during the invasion.”

A complete background of historical data of the snow crab is limited or well not known for the study area,

Reply: The scope of this paper is not the study of population dynamics of the crab in the bay, and not dedicated to show that the crab is the main reason for observed difference. This was shown in the paper cited multiple times in the MS  (Udalov, A.A.; Anisimov, I.M.; Basin, A.B.; Borisenko, G.V.; Galkin, S.V.; Syomin, V.L.; Shchuka, S.A.; Simakov, M.I.; Zalota A.K.; Chikina, M.V. Changes in benthic communities in Blagopoluchiya Bay (Novaya Zemlya, Kara Sea): The influence of the snow crab. Biological Invasions 2024. https://doi.org/10.1007/s10530-024-03388-1) where the crabs data on size and density is given. Here, we aim to look at one of the aspects of changes – trophic – not to argue that the crab is the cause. However, the general dynamics of crabs population is important in this work, therefore the sizes of the crabs at different years are given in the body of the manuscript with the reference to Udalov et al, at multiple times.

Lines 73-85 “Snow crabs settled in Blagopoluchiya Bay in 2014 [13]. In 2016, a large number of 73 small (carapace width CW ~15 mm) juvenile crabs were caught in trawl samples, and this 74 single cohort grew throughout the years without any new settlements [13,22,27]. By 2018, 75 crabs grew (CW ~30 mm), and their abundance was more than 20 crabs per 10 m2. In 76 2020, their number crashed (less than ten crabs per 10 m2) and further decreased in 2022 77 [13]. 78

Lines 73-77 “Snow crabs settled in Blagopoluchiya Bay in 2014 [13]. In 2016, a large number of 73 small (carapace width CW ~15 mm) juvenile crabs were caught in trawl samples, and this 74 single cohort grew throughout the years without any new settlements [13,22,27]. By 2018, 75 crabs grew (CW ~30 mm), and their abundance was more than 20 crabs per 10 m2. In 76 2020, their number crashed (less than ten crabs per 10 m2) and further decreased in 2022 77 [13]. 78

A food web and food pyramid must be outline including the snow crab,

Reply: The food web in Blagopoluchiya Bay is not pyramid shaped. However some representation of the respective positions of different species is given in figure 3 and 4 which do include crab.

Benthic community could change with the recent drastic change in the marine environment including abiotic factors, rather than the snow crab alone,

Reply:  Indeed they could have. But in our recent work we have shown that this is not the case, and crab is the primary cause for the observed changes. The aim of this work is not to prove that the crab has caused the changes, but how the trophic structure has altered during these changes. This is discussed in the introduction with the reference to the paper dedicated to this issue (Udalov, A.A.; Anisimov, I.M.; Basin, A.B.; Borisenko, G.V.; Galkin, S.V.; Syomin, V.L.; Shchuka, S.A.; Simakov, M.I.; Zalota A.K.; Chikina, M.V. Changes in benthic communities in Blagopoluchiya Bay (Novaya Zemlya, Kara Sea): The influence of the snow crab. Biological Invasions 2024. https://doi.org/10.1007/s10530-024-03388-1).

Line 79 and further: “Blagopoluchiya Bay's benthic communities were studied before and during the in-vasion [13,28]. The changes in benthic communities occurred at various levels and were first observed after 2018 when the crab reached a large abundance and size that could feed on larger prey such as molluscs and brittle stars. While changes in macrobenthos were relatively weak, changes in megabenthos were dramatic. By 2022, not a single specimen of previously dominant ophiuroids was found in the bay [13]. No consistent trends in abiotic parameters, such as ice melting, turbidity, temperature, and nutrients were observed during the observation period in the north-western part of the sea along the coast of the Novaya Zemlya Archipelago and Blagopoluchiya Bay [Udalov et al. 2024]. It has been shown that these changes in benthic communities of the bay cannot be attributed to environmental factors; they are the result of the establishment and development of the snow crab population [Udalov et al., 2024].

The MS presentation could be considered a report without the trophic indices and statistical analyses,

Reply: we are not sure what is meant by trophic indices. We have added statistical analysis of trophic position and alpha sensitivity using different base options. The trophic positions is calculated using proven and widely used method described by Post (2002).

Benthic community is influenced easily by the anthropogenic factors inducing also hydrographic variables in a year,

Reply:  indeed benthic communities in areas with high human presence and fishing industry are strongly affected by human operations. However Blagopoluchiya bay is not one such place. This is discussed in the introduction and is shown in the paper dedicated to identifying the chause of changes (which is not the aim of this manuscript): Udalov, A.A.; Anisimov, I.M.; Basin, A.B.; Borisenko, G.V.; Galkin, S.V.; Syomin, V.L.; Shchuka, S.A.; Simakov, M.I.; Zalota A.K.; Chikina, M.V. Changes in benthic communities in Blagopoluchiya Bay (Novaya Zemlya, Kara Sea): The influence of the snow crab. Biological Invasions 2024. https://doi.org/10.1007/s10530-024-03388-1

Food content of the snow crab is missing for the present study,

Reply: The aim of this study in not detailed study of the crabs feeding habits in Blagopoluchiya or any other species. Just the changes in trophic structure based on their trophic positions. We have recently published a paper dedicated to crabs feeding in Blagopoluchiya that includes stomach content analysis, which is cited in this manuscript: Kiselev, A.D.; Zalota, A.K. Changes in the Diet of an Invasive Predatory Crab, Chionoecetes opilio, in the Degrading Benthic Community of an Arctic Fjord. Biology 2024, 13, 781. https://doi.org/10.3390/biology13100781

Energy flow in math and metabolic rates (excretion, assimilation, absorption rates, etc) are missing for the crab,

Reply: We agree that this kind of information is very interesting and would greatly increase the understanding of how crab survives in Blagopoluchiya bay, however this is not the aim or scope of this study. We do indeed plan to work on this issue in the future.

For the stability purposed in the MS, a balanced model could need for input and output

Reply: mass balance model is indeed our aim in the long run, however not the aim of this particular paper. To design correct and accurate model, a good understanding of species composition, their mortality and recruitment, their energy consumption and the sources of this energy is needed. Together with the two papers mentioned above (Kiselev, Zalota 2024; Udalov et al 2024) this paper are steps towards that model. However at present time the trophic position of species in the bay was not known. In addition we did not know if it was changing as the species composition changed. The next step is to study and understand who eats whom exactly and how they could switch. Additional experimental and general information is needed about the metabolic rates to calculate the energy flow. Which we hope will be the next step in our research.

Round 2

Reviewer 2 Report

Comments and Suggestions for Authors

The manuscript (MS) is now acceptable with its weakness and strengthness.  The authors added some more literature and revised the MS better with the modified figures.  Since the authors put a lot effort to design the study and recently introduced species, snow crab, the MS could be baseline study for such polar regions in the future studies.